# Effects of One-Day Application of Levodopa/Carbidopa/Entacapone versus Levodopa/Carbidopa/Opicapone in Parkinson’s Disease Patients

**DOI:** 10.3390/cells11091511

**Published:** 2022-04-30

**Authors:** Thomas Müller, Eugen Schlegel, Stephanie Zingler, Hans Michael Thiede

**Affiliations:** 1Department of Neurology, St. Joseph Hospital Berlin-Weissensee, Gartenstr. 1, 13088 Berlin, Germany; 2Faculty of Medicine, Ruhr University of Bochum, Universitätsstraße 150, 44801 Bochum, Germany; 3Department of Neurology, Kreisklinikum Siegen, Weidenauer Str. 76, 57076 Siegen, Germany; schlegel@zns-siegen.de; 4Swiss Pharm Audit GmbH, Rösliweg 11, 8404 Winterthur, Switzerland; stephanie.zingler@swisspharmaudit.com; 5Thanares GmbH, Hermannswerder 14, 14473 Potsdam, Germany; m.thiede@thanares.de

**Keywords:** opicapone, entacapone, homocysteine, levodopa, 3-O-methyldopa, Parkinson’s disease

## Abstract

The catechol-O-methyltransferase inhibitors entacapone and opicapone prolong the efficacy of conventional oral levodopa/dopa decarboxylase inhibitor formulations through an increase in levodopa plasma bioavailability. Catechol-O-methyltransferase inhibitors influence the homocysteine metabolism associated with levodopa/dopa decarboxylase application. The objectives of this study were to compare the impact of additional single-day entacapone or opicapone intake on the pharmacokinetic plasma behaviour of levodopa, 3-O-methyldopa and total homocysteine in 15 Parkinson’s disease patients, with concomitant scoring of motor symptoms, under standardized conditions. The patients received opicapone plus two doses of 100 mg levodopa/carbidopa and, one week later, two doses of levodopa/carbidopa/entacapone or vice versa. Levodopa, 3-O-methyldopa and total homocysteine were determined with reversed-phase high-performance liquid chromatography. Levodopa bioavailability and its maximum concentration were higher with opicapone. The computed peak-to-trough difference was lower after the second levodopa administration with entacapone. The fluctuation index of levodopa did not differ between both conditions. 3-O-methyldopa decreased on both days. Homocysteine levels did not significantly vary between both conditions. A significant homocysteine decrease occurred with entacapone, but not with opicapone. Motor behaviour improved with entacapone, but not with opicapone. Opicapone baseline scores were significantly better, and thus the potential for the improvement in motor symptoms was lower compared with the entacapone condition. The higher levodopa bioavailability with opicapone suggests that it is more efficacious than entacapone for the amelioration of “off” phenomena in fluctuating patients when co-administered with a levodopa/dopa decarboxylase inhibitor regimen. Both compounds prevented an increase in homocysteine, which is a metabolic marker for an impaired capacity in the performance of methylation processes.

## 1. Introduction

An efficacious, well-tolerated treatment for patients with Parkinson’s disease (PD) is oral levodopa (L-dopa) application combined with an aromatic amino acid decarboxylase inhibitor (DDI) [1]. The plasma L-dopa half-life is approximately 60 to 90 min. This rapid metabolism in combination with intermittent oral intake causes peaks and troughs of peripheral L-dopa plasma levels. This variability is further promoted by gastrointestinal transport and absorption of L-dopa [2]. One approach for prolonging the efficacy of each L-dopa/DDI dose, smoothing out the fluctuations in L-dopa plasma levels and enabling a more continuous L-dopa delivery to the brain is the application of retarded-release L-dopa/DDI formulations [3]. Another is the peripheral enzyme inhibition of catechol-O-methyltransferase (COMT-I) with opicapone (OP) or entacapone (EN) [4,5]. OP with its once-daily application regimen provides a more constant COMT-I compared to EN with its shorter half-life and the greater fluctuations through COMT enzyme blocking that is associated with it [6,7]. Therefore, EN should be applied with each oral L-dopa/DDI formulation [8]. There is some evidence from pharmacokinetic studies that the fall in L-dopa levels in plasma is less pronounced with COMT-I, i.e., during a dosing interval of every 3 h [9,10]. To date, no pharmacokinetic trial has compared the ups and downs of L-dopa plasma levels following repeat conventional L-dopa/DDI application with the COMT-I’s EN and OP during a dosing interval of 4 h under standardised conditions. Moreover, no study in PD patients has performed a head-to-head comparison between EN and OP concerning the synthesis of the L-dopa metabolite 3-O-methyldopa (3-OMD) in PD patients. 3-OMD competes with L-dopa near the large neutral amino acid transport carriers at the blood brain barrier. Thus, high 3-OMD levels during L-dopa/DDI administration may contribute to a reduced L-dopa delivery to the brain, in particular during repeat administration. By contrast, COMT inhibition decreases 3-OMD synthesis and therefore improves the blood brain barrier transfer of L-dopa. Thus, high 3-OMD levels may additionally weaken the motor response to L-dopa [11]. Treatment with L-dopa/DDI also impairs methylation capacity, which is mirrored by a total homocysteine (thcys) elevation in plasma [12,13,14]. No study has investigated how OP and EN influence thcys metabolism in the same cohort of PD patients. The objectives of this study were to compare the impact of EN or OP intake on the pharmacokinetic behaviour of L-dopa, 3-OMD and thcys in PD patients, with concomitant scoring of motor symptoms.

## 2. Subjects and Methods

### 2.1. Subjects

COMT-I-naïve Caucasian PD out-patients participated in this trial. One dropped out, as the inclusion criteria were not fulfilled. Another did not perform the second condition because of an infection. The Eppendorf caps of three participants were not correctly labelled. Therefore, the identification of the frozen samples was faulty before or following unfreezing. Accordingly, 15 PD participants were analysed. Their clinical data are reported in a table below. The data collection was performed 7 days before day 1 during a screening visit. The inclusion criteria included a diagnosis of PD according to the UK Brain Bank, age > 50 years, a duration of PD > 1 year, Hoehn and Yahr Stage I-IV, a Unified Parkinson’s Disease Rating Scale (UPDRS) mental behaviour (I) > 0, UPDRS activities of daily living (II) > 2, UPDRS motor examination (III) > 4, UPDRS complications of therapy (IV) > 0. Patients using dopamine-substituting pump devices or undergoing deep brain stimulation were not allowed to participate. Further exclusion criteria were exposure to neuroleptics or any other drugs which aggravate motor symptoms in PD patients, clinical signs of dementia, severe comorbidities, hypersensitivity to the active substance or to any of the excipients, a history of phaeochromocytoma, paraganglioma, or other catecholamine secreting neoplasms, a history of neuroleptic malignant syndrome and/or non-traumatic rhabdomyolysis, and concomitant use of monoamine oxidase (MAO-A and MAO-B) inhibitors other than those approved for the treatment of PD. 

### 2.2. Design

UPDRS III scoring was performed with blinded raters in this randomised, cross-over trial on both investigation days. A total of 7 PD patients received OP on day 1, 8 PD patients took L-dopa/carbidopa/entacapone (Stalevo^®^, LD/CD/EN) on day 1. There was an interval of 7 days between both investigation days. The participants were on their normal PD drug regimen during this period. PD drugs other than the study drugs were not taken on investigations days 1 and 2 until the end of the period with blood sampling. Then, the participants took their regular medication. A total of 50 mg OP was administered at 7.30 am and then 100 mg L-dopa/CD at 8 am and 12 pm (LD/CD/OP). A total of 100 mg LD/CD/EN was applied at 8 am and 12 pm. Standardized meal intake was at 8.30 am and 12.30 pm. The food was the same on both investigation days. No high-fat meals were provided. Blood sampling was performed at 7.30, 8.30 (before the meal), 9, 10, 11, 12 (before drug application), 1, 2, 3, 4 (am and pm, respectively). Scoring with the UPDRS III was performed at 7.30, 9, 10, 11, 12 (before drug administration), 1, 2, 3, 4 (am and pm, respectively) by UPDRS-certified physicians [15]. 

### 2.3. Statistical Analysis

The parameters, total area under the curve (AUC) and the maximum and minimum concentrations (C_max_, C_min_), respectively, were calculated using the linear trapezoidal rule over the interval (Table 1). If applicable, comparisons were performed by paired t-tests, and ANOVA, respectively, with a repeated measures design. Tukey’s multiple comparison test was employed for the ANOVA post hoc analysis. Parametric tests were chosen, since both ordinal and metric biological data were analysed. Covariates were not set, since calculations were performed for the same patient cohort. The peak-to-trough difference (C_max_-C_min_) was computed for the periods after the first (30–240 (interval in minutes after baseline)) and second intake (270–480) of L-dopa/DDI [16]. The fluctuation index (FI) was calculated with the formula C_max_-C_min_/C_average_ for the first (0–240), second (240–480) and whole interval (0–480) [17]. Pearson’s product correlation was used for correlation analysis. The significance of the *p*-level was set to *p* < 0.05 in this exploratory, descriptive analysis. 

### 2.4. Blood Sampling

A total of 10 mL venous blood was needed for the estimation of L-dopa and 3-OMD plasma levels. Blood was taken from an antecubital vein through an indwelling catheter. A venous puncture was performed 20 min before the baseline investigation to enable stable conditions. Then, 3 mL of blood was drawn with a separate syringe and discarded before taking each 10 mL specimen. The blood samples were collected in EDTA test tubes containing 100 µ of 0.5% sodium disulphite solution. The plasma obtained from rapid centrifugation was immediately frozen at −80 °C until analysis. Reversed-phase high-performance liquid chromatography was employed in combination with electrochemical detection for the measurement of L-dopa, 3-O-methyldopa (3-OMD) and thcys levels in the plasma [18].

### 2.5. Ethics 

All participants provided written informed consent. The study was approved by the local ethics committee of the Ruhr University of Bochum (17-6250-AMG-FF/Protocoll Nr: PD-08-2017; EUDRA Number: 2017-003273-34). The committee restricted the number of blood samples to 10 per day and the frequency of motor behaviour evaluations to 7 per day. They did not allow a total overnight withdrawal of all PD drugs, formerly taken on a regular basis, with the exception of the intake of L-dopa/DDI. The committee argued to ensure a safer trip on the two investigation days. 

## 3. Results

### 3.1. L-dopa

The raw values, as well as AUC and C_max_, were higher during LD/CD/OP than during LD/CD/EN application (Table 2, Figure 1). L-dopa levels increased with LD/CD/OP (ANOVA F = 8.86, *p* < 0.0001, post hoc analysis, see Figure 1). L-dopa plasma concentrations were higher with COMT-I and OP compared to EN at moments 30 (*p* = 0.031) and 300 (*p* = 0.016) (Figure 1). L-dopa levels also increased during LD/CD/EN application (ANOVA F = 6.17, *p* < 0.0001) (post hoc analysis, see Figure 1). The computed peak-to-trough difference was higher with LD/CD/OP than with LD/CD/EN after the intake of the second drug, but not after the first application (Figure 1, Table 2) [16]. The FI did not differ (Table 2). 

### 3.2. OMD in Plasma

AUC and C_max_ did not differ between both conditions (Table 2). Significant reductions were found (LD/CD/OP: ANOVA F = 5.33, *p* < 0.0001; LD/CD/EN: ANOVA F = 5.43, *p* < 0.0001) (Figure 2). The post hoc analysis was negative. No differences appeared between both conditions when each assessment moment was compared with the corresponding one for the other condition.

### 3.3. Thcys in Plasma

The AUC and C_max_ values as well as the concentrations at each moment did not differ between both investigation days (Table 2, Figure 3). Thycs decreased during LD/CD/EN application (ANOVA F = 2.81, *p* < 0.0001, post hoc analysis: moment 0 vs. 420: *p* < 0.01), but not during LD/CD/OP application (ANOVA F = 1.93, *p* = 0.054). There was a weak correlation between the computed differences in AUC 3-OMD and AUC thcys between LD/CD/OP and LD/CD/EN (R = −0.56, *p* = 0.03). 

### 3.4. UPDRS III

No differences in the AUC and C_min_ values appeared (Table 2, Figure 4). UPDRS III scores decreased during LD/CD/EN application (ANOVA F = 5.21, *p* = 0.0002; post hoc analysis, see Figure 4), but not during LD/CD/OP application (ANOVA F = 1.11, *p* = 0.36). The UPDRS III values at moment 0 were lower during LD/CD/OP application (*p* = 0.045) (Figure 4).

Body weight, computed body mass index and the condition sequence (EN initially and then OP or vice versa) did not influence the results.

## 4. Discussion

COMT-I with OP provided higher L-dopa plasma levels compared with EN. Accordingly, C_max_ and the peak-to-trough values were higher during LD/CD/OP after the second LD/CD intake [16]. This elevated L-dopa plasma bioavailability observed with OP intake hypothetically suggests that OP addition to L-dopa/DDI is more efficacious and will probably better improve “off” times compared with LD/CD/EN in fluctuating LD/CD/EN- or LD/CD-treated PD patients in the long term [5,19]. However, this L-dopa plasma increase may also hypothetically contribute to peak dose dyskinesia onset. It is known from pharmacokinetic trials that repeat EN dosing, i.e., every 3 h, increases the C_max_ and C_min_ of L-dopa in plasma. Accordingly, a less pronounced fall in L-dopa occurred during COMT-I administration [9,10,17]. L-dopa-naïve patients received oral L-dopa/DDI formulations only three times per day in the FIRST-STEP and in the ELLDOPA trials. Therefore, a plasma accumulation of L-dopa was more unlikely and, if it occurred at all, after the administration of higher L-dopa dosages only. Accordingly, dyskinesia more rarely appeared and LD/CD/EN was not inferior to traditional LD/CD formulations in terms of the observed frequency of dyskinesia in both studies [1,20]. However, this interplay between pharmacokinetic plasma behaviour, dosing intervals and COMT-I could be one reason for the observed premature appearance of dyskinesia in the LD/CD/EN arm of the STRIDE-PD trial with its mandatory L-dopa dosing increase as a result of a more pronounced L-dopa plasma accumulation [21]. In our present study, with a dosing interval lasting 4 h, the continuity of L-dopa plasma behaviour and the associated brain delivery was similar with both forms of COMT-I, as the computed fluctuation indices did not differ. 

3-OMD plasma levels did not vary between LD/CD/OP and LD/CD/EN, as both compounds inhibited the O-methylation of L-dopa. They decreased 3-OMD enrichment to a nearly equal extent. 3-OMD has a long half-life of approximately 24 h. Nevertheless, both COMT-I forms were potent enough to reduce 3-OMD levels even during this short investigation period [2]. The weak association between the differences in AUC 3-OMD and AUC thcys underlines the metabolic relationship between thcys and L-dopa metabolism in the periphery [2].

A fall in thcys appeared during LD/CD/EN application, but no significant decrease was found during LD/CD/OP application. Both outcomes confirm that COMT-I prevents thcys elevation, in contrast to L-dopa/DDI application only [13,14,22]. The four-hour time period between the two L-dopa/DDI applications was probably responsible for the initial observation of a weak thcys fall due to COMT-I, in contrast to earlier investigations with a 3 h interval between L-dopa/DDI administrations [14,22]. A thcys increase is reflected as a biomarker of an impaired methylation potential. It may accelerate ageing processes in the central and peripheral nervous system in the long term [23]. The missing significance of the thcys fall during LD/CD/OP probably resulted from the higher bioavailability of L-dopa in comparison to LD/CD/EN. It may hypothetically indicate that only a certain metabolic capacity is available for the compensation of methyl group consumption during L-dopa metabolism to 3-OMD.

LD/CD/EN but not LD/CD/OP significantly reduced the severity of motor impairment. No further relevant differences were found. One must consider that UPDRS III scores at moment 0 were significant higher during the LD/CD/EN condition. This difference may explain this outcome to a certain extent, as the potential for motor symptom improvement was lower during LD/CD/OP compared with LD/CD/EN application. 

Some limitations of this trial are the relatively low frequency of the plasma determinations of L-dopa, 3-OMD, thcys and the ratings with the UPDRS III due to the ethical requirements and the number of participants in general. OP was administered 30 min before LD/CD intake and not on the evening before as in clinical practice due to the performance of the trial with out-patients. Thus, we were able to control the OP intake. Only one female PD patient was included in the analysis; therefore, this trial does not allow the generalization of these results to the whole population. 

In conclusion, we showed that OP is superior to EN in terms of L-dopa plasma bioavailability, which supports the assumed stronger clinical effect of OP compared to EN [24]. 3-OMD plasma behaviour did not differ. Both COMT-I forms prevented an increase in thcys, which reflects the methylation capacity with its impact on numerous tasks in the human body, such as DNA methylation or detoxification processes via N-methylation [25]. This metabolic advantage hypothetically suggests the initiation of L-dopa therapy with dual enzymatic DDI and COMT-I, if tolerated. We stress that there is a need to confirm the probable beneficial clinical consequences of a prevention of L-dopa/DDI-induced thcys elevation in a long-term study [2].

## Figures and Tables

**Figure 1 cells-11-01511-f001:**
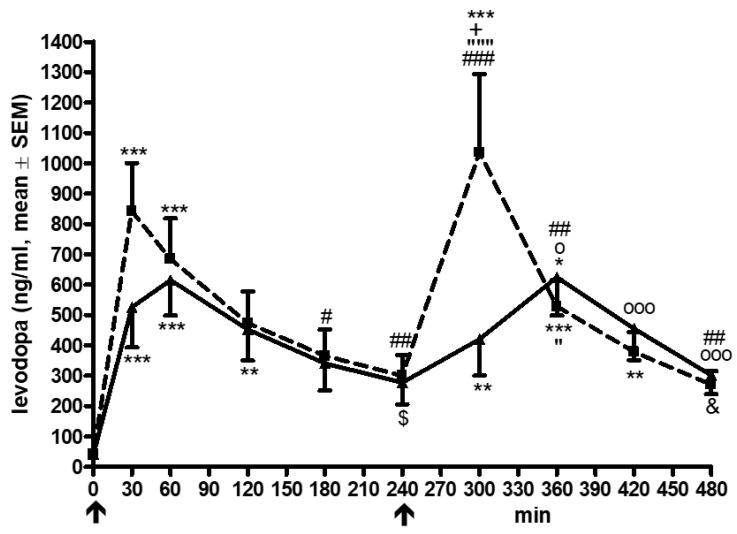
Curve of L-dopa plasma concentrations. All data are given as mean ± standard error of means (SEM); min, minutes; ↑ marks the L-dopa/DDI intake at moment 0 and after 240 min; opicapone was taken at −30 min; L-dopa plasma levels are given in ng/mL, thcys concentrations in µmol/L; lines of the L-dopa/carbidopa plus opicapone condition are dashed with ■; lines of the L-dopa/carbidopa/entacapone condition are continuous with ▲; comparisons of the post hoc analysis: *, *p* < 0.05, ** *p* < 0.01, *** *p* < 0.001; * reports comparisons against moment 0; #, *p* < 0.05, ## *p* < 0.01, ### *p* < 0.001; # reports comparisons against moment 30; $, *p* < 0.05; $ reports comparisons against moment 60; + *p* < 0.05, + reports comparisons against moment 120; &, *p* < 0.05; & reports comparisons against moment 180; “, *p* < 0.05; “““, *p* < 0.001; “ reports comparisons against moment 240; o, *p* < 0.05; ooo, *p* < 0.001; o reports comparisons against moment 300; symbols for the post hoc analysis and error bars are above the curve of the L-dopa/carbidopa plus opicapone condition; symbols for the post hoc analysis and error bars are below the curve of the L-dopa/carbidopa/entacapone condition.

**Figure 2 cells-11-01511-f002:**
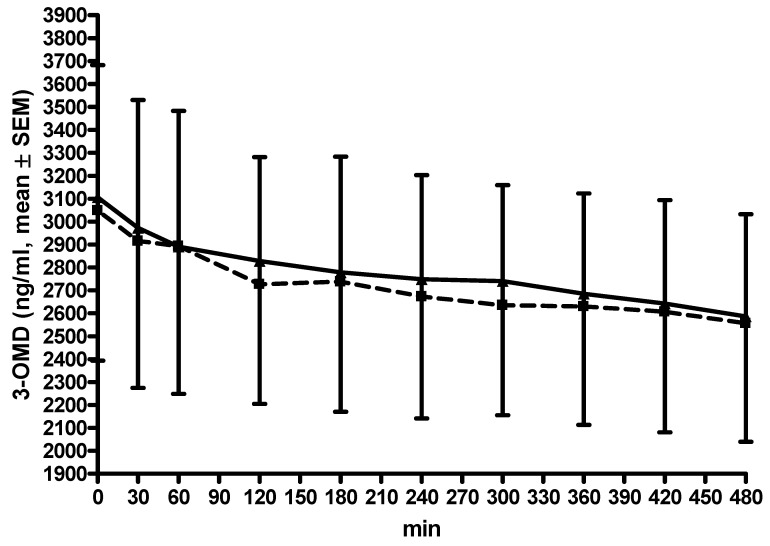
Curve of 3-OMD levels. All data are given as mean ± standard error of means (SEM); min, minutes; opicapone was taken at −30 min; L-dopa and 3-OMD plasma levels are given in ng/mL, thcys concentrations in µmol/L; lines of the L-dopa/carbidopa plus opicapone condition are dashed with ■; lines of the L-dopa/carbidopa/entacapone condition are continuous with ▲; error bars are above the curve of the L-dopa/carbidopa plus opicapone condition; error bars are below the curve of the L-dopa/carbidopa/entacapone condition.

**Figure 3 cells-11-01511-f003:**
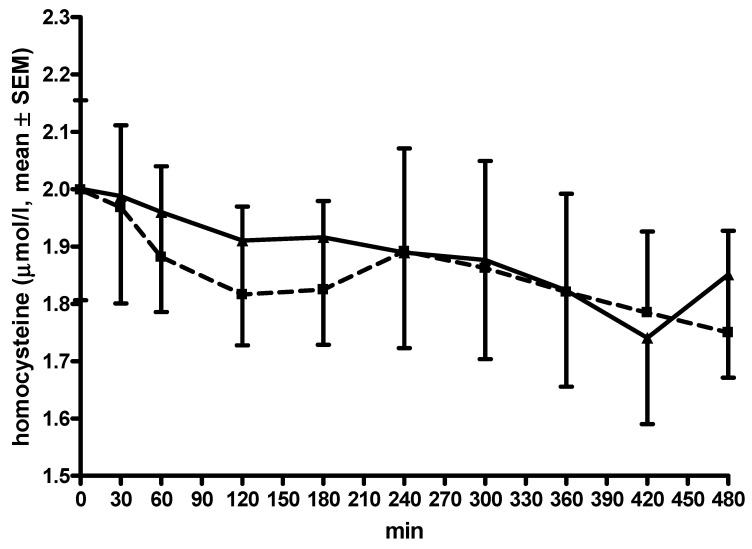
Curve of total homocysteine. All data are given as mean ± standard error of means (SEM); min, minutes; opicapone was taken at −30 min; thcys concentrations in µmol/L; lines of the L-dopa/carbidopa plus opicapone condition are dashed with ■; lines of the L-dopa/carbidopa/entacapone condition are continuous with ▲; error bars are above the curve of the L-dopa/carbidopa plus opicapone condition; error bars are below the curve of the L-dopa/carbidopa/entacapone condition.

**Figure 4 cells-11-01511-f004:**
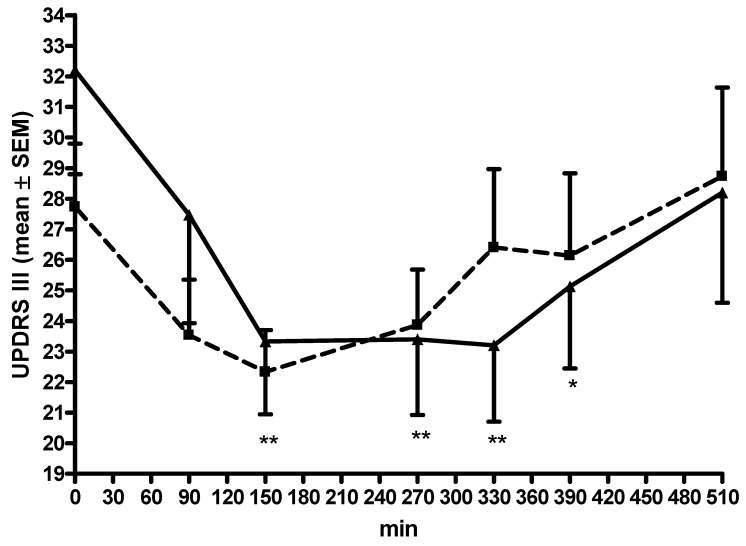
Scoring outcomes over the interval. All data are given as mean ± standard error of means (SEM); min, minutes; UPDRS III, Unified Parkinson’s Disease Rating Scale for motor examination; lines of the L-dopa/carbidopa plus opicapone condition are dashed with ■; lines of the L-dopa/carbidopa/entacapone condition are continuous with ▲; comparisons of the post hoc analysis: * *p* < 0.05, ** *p* < 0.01; * reports comparisons against moment 0; error bars are above the curve of the L-dopa/carbidopa plus opicapone condition; error bars are below the curve of the L-dopa/carbidopa/entacapone condition.

**Table 1 cells-11-01511-t001:** Clinical data of the patient group. All data are given as mean ± standard deviation. UPDRS I, Unified Parkinson’s Disease Rating Scale mental behaviour; UPDRS II, UPDRS activities of daily living; UPDRS III, UPDRS motor examination; UPDRS IV, UPDRS motor complications; UPDRS total, sum of UPDRS I, II, III and IV.

Age	64.67 ± 8.47 years
Sex	14 men, 1 woman
Time since diagnosis	10 ± 3.58 years
Height	178.6 ± 10.2 cm
Body weight	92.56 ± 14.67 kg
UPDRS I	2.8 ± 1.59
UPDRS II	13.8 ± 4.83
UPDRS III	31.6 ± 9.63
UPDRS IV	4.8 ± 1.18
UPDRS total	52.8 ± 12.87
Daily oral L-dopa/carbidopa (benserazide) dose	444.4 ± 133.5 mg
Safinamide 100 mg	2 patients
Selegiline 5 mg	1 patient
Rasagiline 1 mg	4 patients
Piribedil	2 patients 150 mg, 1 patient 50 mg
Pramipexol	4 patients 3.15 mg, 1 patient 1.57 mg
Rotigotine	2 patients 8 mg, 1 patient 6 mg
Ropinirole	2 patients 4 mg, 1 patient 8 mg
Amantadine	3 patients 200 mg

**Table 2 cells-11-01511-t002:** Comparison between both conditions. AUC, area under the curve; AUC UPDRS III; number * 510 min, AUC thcys µmol/L * 480 min, AUC 3-OMD, -L-dopa ŋg/mL * 480 min; C_max_, maximum concentration; C_min_, minimum concentration; C_minUPDRS_ lowest UPRDS III score; ns, not significant; data are given as mean ± SEM; 1st period: C_max_-C_min_; 2nd period: C_max_-C_min_; peak-to-trough calculations as defined in the Methods section; FI, fluctuation index; FI: 1st interval; FI: 2nd interval; FI: whole interval as defined in the Methods section; *p*, *p*-value; L-dopa and 3-OMD plasma levels are given in ŋg/mL, thcys concentrations in µmol/L; UPDRS III, Unified Parkinson’s Disease Rating Scale for motor examination.

	LD/CD/OP	LD/CD/EN	
	L-dopa		*p*
Raw values L-dopa	511.3 ± 46.25	405.7 ± 33.56	0.0029
AUC	249,810 ± 39,025	207,467 ± 39,191	0.0014
C_max_	1370 ± 235.5	910.3 ± 142.5	0.0063
1st period: C_max_-C_min_	628.1 ± 108.8	495.1 ± 89.14	ns
2nd period: C_max_-C_min_	820.3 ± 223.9	333.6 ± 67.43	0.019
FI: 1st interval	1.77 ± 0.11	1.96 ± 0.09	ns
FI: 2nd interval	1.56 ± 0.24	1.15 ± 0.18	ns
FI: whole interval	2.56 ± 0.24	2.36 ± 0.22	ns
	3-OMD		
AUC	1.301e + 006 ± 256,335	1.329e + 006 ± 290,178	ns
C_max_	3151 ± 612.9	3219 ± 690.7	ns
	thcys		
AUC	886.2 ± 74.84	902.8 ± 81.11	ns
C_max_	2.213 ± 0.1921	2.169 ± 0.1935	ns
	UPDRS III		
AUC	12,831 ± 938.6	13,061 ± 1321	ns
C_minUPDRS_	19.33 ± 1.472	20 ± 2.348	ns

## Data Availability

The data presented in this study are available on request from the corresponding author.

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
