# Peer review of "Effects of One-Day Application of Levodopa/Carbidopa/Entacapone versus Levodopa/Carbidopa/Opicapone in Parkinson’s Disease Patients"

_cells, 2022, doi:10.3390/cells11091511_

Round 1

Reviewer 1 Report

  1. More details/specifications about the rationale behind the study design!
  2. Dilemmas: is morning administration of opicapone relevant for proper pharmacokinetics? More information must be provided about PD add-on medication (particularly rasagiline and safinamide if any)? Comorbidities and corresponding medication? Any dietary restriction -if any, must be specified (high-fat meals! could delay the time of peak plasma level)? Hepatic functions? (liver impairment increase opicapone level)
  3. Discuss the homogeneity of the patient population
  4. Precise how the patients continued their regular medication after the last pharmacokinetic evaluation.

Reviewer 2 Report

This is a well written article, I have only a few questions and suggestions :

  1. the mean body weight of the group is high, atypical for  PD patients, and the results were not influenced by that. My question is if the authors found any remarkably different individual result  related to body weight. This can be important when planning further similar studies.
  2. a similar question for UPDRS, motor and total:  maybe clinicians would be interested if there were particular or unexpected changes of the scores for some patients after OP or EN.
  3. row 265: truncated text, probably: "stronger clinical effect of OP"

Reviewer 3 Report

On this study, Müller and collegues compared the impact of 2 catechol-O-methyl transferase Inhibitors, entacapone (EN) and opicapone (OP), in Parkinson disease patients. They analyzed the pharmacokinetic profile of L-dopa, 3-O-methyldopa and total homocysteine by HPLC and they scored the motor symptoms using UPDRS scale. They showed that OP intake results in a higher L-DOPA bioavailability. Both compounds are effective preventing the increase of homocysteine. The results are interesting, however the study has some limitations, and several issues must be addressed:

  • The results are obtained from a small sample (N=15, 8EP +7OP), 14 men and just 1 woman. Clinical results are often subjected to high variability, and it is important to get a larger sample size and a balanced number between women and men.
  • The center/hospital from which the subjects come is not specified in the methods section. Multicentric studies are more valuable in clinical studies
  • Experimental design: protocol of administration of EN and OP seems different (a scheme or timeline of the design protocol, showing the administration pattern and the different time points of sample analysis would be useful). Please, explain the reasons of that choice, in a pharmacokinetic study the pattern of administration of both compounds should be the same. OP should be administered concomitantly with L-Dopa/CD, just like EN, earlier administration of OP will lead to a different blocking level of COMT, i.e. OP will block COMT before L-Dopa gets metabolized, while EN will block COMT at the same time L-Dopa gets metabolized, thus, this may lead to higher L-dopa metabolization. It would be convenient to prolong the study along the time, when plasmatic levels are more stable. Moreover, It would be useful a scheme or timeline of the design protocol, showing the administration pattern and the different time points of sample analysis
  • In the abstract the authors mentioned: “Motor behaviour improved with entacapone, but not with opicapone”, however in the results no differences of the AUC and Cmin values appeared in the UPRDS III scores, please clarify this issue.
  • The authors suggested that increased plasma levels of L-dopa after administration of OP could contribute to dyskinesia onset, however they do not analyse dyskinetic behaviour on the patients. Dyskinesia is one of the most complication after chronic L-dopa treatment, then it is important pay attention of that point.
  • Methods section should describe the protocol and details of HPLC techniques  
  • In the figures 1, 2 and 3 the graphs neither the figure legends do not specify which line/symbol corresponds to each treatment.
  • An interesting review was published last year about the use of different Catechol-O-Methyl Transferase Inhibitors in patients:

Different Catechol-O-Methyl Transferase Inhibitors in Parkinson's Disease: A Bayesian Network Meta-Analysis. Song Z, Zhang J, Xue T, Yang Y, Wu D, Chen Z, You W, Wang Z. Front Neurol. 2021. doi: 10.3389/fneur.2021.707723. eCollection 2021

Minor points:

Lines 193 and 209: abbreviature of OP and EN should be used by coherence

Please, check the use of hyphens:

Line 73, no need to add the hyphen after EN in EN- or OP intake

Line 74, the abbreviation for total homocysteine has been stablished as thcys, not t-hcys

Line 189, no need for the hyphen after AUC

Line 223, no need for the hyphen after STEP (STEP-) 

Line 239, no need for the hyphen after thcys (thcys -) 

 Line 24, 63, 109 correct standardized by standardized

Author Response

please see the attachmend

Reviewer 4 Report

Cell-1661940

 Thank you for the opportunity to review the manuscript entitled

Effects of one day application of levodopa/carbidopa/entacapone versus levodopa/carbidopa/opicapone in Parkinson’s disease patients

This is a very interesting trial and very helpful for the movement disorders specialists, however, I have some comments to the authors.

Could you please clarify in the inclusion criteria the cut-off for the UPDRS? It is  confuse. For instance, for UPDRS mental behavior I: > 0, motor examination III > 4 meaning that any patient could fit this inclusion criteria.   Also, the nomination of each trace and the important moments in the figures should be described below of each graph. It makes easier to understand.

Round 2

Reviewer 3 Report

The authors solved more and less the issues requesed in the first review. My recommendation is"accept in present form"